# Entropy-Based Temporal Downscaling of Precipitation as Tool for Sediment Delivery Ratio Assessment

**DOI:** 10.3390/e23121615

**Published:** 2021-12-01

**Authors:** Pedro Henrique Lima Alencar, Eva Nora Paton, José Carlos de Araújo

**Affiliations:** 1Institut für Ökologie, Technische Universität Berlin, Ernst-Reuter-Platz 1, 10587 Berlin, Germany; eva.paton@tu-berlin.de; 2Departamento de Engenharia Agrícola, Campus do Pici Fortaleza, Universidade Federal do Ceará, Fortaleza 60020-181, Brazil; jcaraujo@ufc.br

**Keywords:** maximum entropy, sediment transport, sediment yield, hydrology

## Abstract

Many regions around the globe are subjected to precipitation-data scarcity that often hinders the capacity of hydrological modeling. The entropy theory and the principle of maximum entropy can help hydrologists to extract useful information from the scarce data available. In this work, we propose a new method to assess sub-daily precipitation features such as duration and intensity based on daily precipitation using the principle of maximum entropy. Particularly in arid and semiarid regions, such sub-daily features are of central importance for modeling sediment transport and deposition. The obtained features were used as input to the SYPoME model (sediment yield using the principle of maximum entropy). The combined method was implemented in seven catchments in Northeast Brazil with drainage areas ranging from 10^−3^ to 10^+2^ km^2^ in assessing sediment yield and delivery ratio. The results show significant improvement when compared with conventional deterministic modeling, with Nash–Sutcliffe efficiency (NSE) of 0.96 and absolute error of 21% for our method against NSE of −4.49 and absolute error of 105% for the deterministic approach.

## 1. Introduction

Climate change challenges our capacity to preserve natural resources, such as clean water and productive soil. The Food and Agriculture Organization named erosion as one of the most relevant threats to soil conservation and agriculture [1]. Climate change is blamed for erosion rates increasing by nearly 17% in the USA and Europe until 2050 due to higher rainfall erosivity [2,3]. This is why soil erosion turned into a key challenge for the Sustainable Development Goals of the UN [4,5]. Soil erosion also imposes a threat to water supply, as pollutants and heavy metals are transported along with sediment, augmenting toxicity, turbidity and eutrophication in aquatic environments [6,7].

In addition, 30% of all land on Earth has an arid or a semiarid climate [8], which causes some places to be especially vulnerable to climate change and soil erosion [9]. Special attention is required for semiarid regions, since they house and sustain over 14% of the global population and around 70% of the dry-land population [9]. Arid and semiarid areas are commonly affected by data scarcity, particularly in Africa, Asia and South America [10,11,12]. It is necessary to improve sedimentological and other models in order to better estimate the amount of sediment reaching water bodies. Modelers normally have information only on daily precipitation data, yet sub-daily processes play a crucial role in sediment transport, as a substantial amount occurs during high-intensity storms [13,14]. Therefore, we need a methodology to downscale precipitation duration and to improve erosion models at the sub-daily scale.

Diverse branches of water sciences point out the use of stochastic methods in hydrology as being the next generation of models [15,16]. In this context, a powerful tool deployed in several studies over the last decades is the principle of maximum entropy (PoME—[17,18]). The first applications of the PoME in water sciences were proposed by Chiu [19] and by Singh and Chowdhury [20] for modeling velocity distribution in open channels. Since then, several other applications in hydrology, hydraulics and sedimentology have been presented [21,22,23,24,25].

de Araújo [26] proposed a PoME-based model to assess sediment yield and reservoir siltation. The model (sediment yield using the principle of maximum entropy—SYPoME), however, requires sub-daily data, such as rainfall duration and intensity measurements, which are often unavailable in arid and semiarid regions [27], such as the Brazilian northeast region. According to the Brazilian Water Management Agency [28], the country’s semiarid region has 2163 operating rainfall stations connected to the national weather monitoring system, which averages one rain gauge per 462 km^2^. Most of those instruments are standard Ville de Paris gauges, providing only daily precipitation. Only 36 are active and reliable automatic stations providing sub-daily precipitation data—one every 27,800 km^2^, on average (Appendix A). The gauging station density is much lower than in other regions (e.g., the density of automatic stations is one per 3600 km^2^ in the United States and 77 km^2^ in Italy—[29,30]). The data series are also not long; only 16 stations have more than 15 years of continuous data.

The Brazilian northeast (10^6^ km^2^) has an average annual temperature varying between 20 and 28 °C and is characterized by a high temporal and spatial rainfall variability [31], with average annual rainfall between 400 mm and 800 mm (increasing towards the coast—[32,33]) and evapotranspiration between 2000 and 2600 mm per year [34]. The vegetation is mainly Caatinga, formed by deciduous broadleaf bushes. The largest part of the region is placed over Precambrian crystalline bedrock with shallow soils. In these areas, groundwater is scarce and usually salty [35,36]. The simultaneous occurrence of such geological features, concentrated precipitation patterns and high evaporation rates leads to a scenario where rivers are predominantly intermittent [37]. As a result, water for over twenty million people living in the Brazilian northeast region is mainly supplied by reservoirs [6]. The region has a concentration of reservoirs as high as one per 5 km^2^ [38]. Due to excessive erosion and eutrophication, however, reservoir siltation is one of the key threats to the water supply in the region [6].

Our objectives are as follows: (1) to propose a temporal down-scaling method to estimate sub-daily precipitation data from daily precipitation data based on the principle of maximum entropy (MEDRID); (2) to assess the method quality when implemented on ungauged regions (spatial-scalability); and (3) to evaluate the effect of the method on the performance of long-term sediment yield modeling.

In order to achieve these objectives, measured data of high-resolution precipitation were used to calibrate and validate the MEDRID method, and the statistical distance measures after Kullback [39] and Fedotov et al. [40] were used to assess spatial scalability. Measured sediment yield data of seven catchments of different sizes and series durations were employed to test and validate the improved sediment yield modeling using scaled precipitation together with the model by de Araújo [26], which is based on entropy equations and quantifies gross erosion by means of the universal soil loss equation (USLE).

## 2. Materials and Methods

Sediment yield can be quantified by multiplying gross erosion and sediment delivery ratio (SDR—[41,42,43]). These terms are highly nonlinear, and deterministic models do not always account for their uncertainties [15,43,44]. Therefore, such processes need to be modeled stochastically and event-wise [15,45]. In this study, the sediment yield of sub-daily events was quantified using the principle of maximum entropy (PoME). To incorporate sub-daily rainfall information, we developed temporal-downscaling equations to assess the effective rainfall duration (*D*) and its respective 30-min intensity ($I_{30}$). As proposed by de Araújo [26], the rainfall duration was drawn on to calculate the SDR, and the $I_{30}$ to calculate the erosivity factor of the universal soil loss equation [46], so as to assess gross erosion.

A new method (Figure 1) was proposed to estimate sediment yield: it consists of an entropy-based approach to downscale rainfall duration and intensity (the MEDRID—maximum entropy distribution of rainfall intensity and duration method). We coupled MEDRID with the SYPoME model to determine an event-wise SDR [26].

### 2.1. Maximum Entropy Distribution of Rainfall Intensity and Duration—MEDRID Method

Two sub-daily variables were selected to be assessed from daily rainfall data: (1) the duration–precipitation ratio D/H (*D* for duration and *H* for total daily precipitation) and (2) intensity–precipitation ratio $I_{30}/H$ (where $I_{30}$ stands for 30-min intensity). Three probability density functions were tested to fit D/H frequencies: the beta (B3), the gamma (G2) and the generalized gamma (G3) distributions [24,47]. For the intensity–precipitation ratio ($I_{30}/H$), two probability density functions were tested: the beta (B3) and the uniform distribution. After calibrating the equations using the principle of maximum entropy [48], we tested the best fitting equations to measured data, as well as spatial scalability.

Table 1 presents the three probability density functions (PDFs—beta, gamma and generalized gamma), their constraints and the respective system of equations for parameterization. Ψ(·) is the digamma function, the first derivative of Γ(·), the gamma function. $\Psi^{\prime }\left(\cdot \right)$ is the tri-gamma function, the second derivative of Γ(·). The terms *a*, *b* and *c* in the three distributions are parameters obtained maximizing entropy using the Lagrange multipliers method [25]. The systems of equations in Table 1 can be solved using empirical data (e.g., rain gauge readings, as for this study—[20]). The parameter *r* in the beta distribution (B3) is a scale factor. For this specific distribution, the random variable X∈[0,1]. The systems of equations were solved with help of the software Octave (v. 5.1.0.0).

Additionally, sub-daily data are scarce and stations may cover a large area. It is important to assess the loss in performance of the method when using data from a distant station. This loss of performance can be measured as the difference between the calibrated PDF for the weather station and the expected PDF, if the region of study had such a station. In this study we compared the variations among four stations with sub-daily data (Aiuaba, Sobral, Sumé and Gilbués) using the Kullback–Leibler divergence [49] and the Kolmogorov–Smirnov distance [50,51]. These statistical measures allow us to find similarities between the areas, and therefore to determine which areas can be modeled with which calibrated PDF without a significant performance loss.

Let *m* and *n* be two populations (sets)—in our study, automatic stations—each with an associated PDF $p_{m}$ and $p_{n}$. Kullback and Leibler [49] present a measure that allows us to compare how different those two distributions are. Known as the Kullback–Leibler divergence, the $D_{KL}$ is an asymmetric measure, given by Equation (Equation 1).
(1)$$D_{KL}\left(P_{m}\|P_{n}\right)=I\left(m:n\right)=\int_{0}^{+\infty }p_{m}\left(x\right)\ln\left[\frac{p_{m}\left(x\right)}{p_{n}\left(x\right)}\right]dx$$
(2)$$J\left(m,n\right)=\frac{I(m:n)+I(n:m)}{2}$$
where $p_{m}$ and $p_{n}$ are continuous probability distributions. I(m:n) can be understood as the loss of information if the population *m* is modeled using $p_{n}$ instead of $p_{m}$. Furthermore, ref. [39] introduces a symmetric measure, given by Equation (Equation 2). J(m,n) is also a measure of divergence between the distributions $p_{m}$ and $p_{n}$ and can be interpreted as how easily we can distinguish the two distributions, henceforth called symmetric divergence.

The Kolmogorov–Smirnov distance (δ—Equation (Equation 3)) is the maximum distance between two distributions in their domain and is related to the Kullback–Leibler divergence by Pinsker’s inequality (Equation (Equation 4)).
(3)$$\delta \left(P_{m},P_{n}\right):=sup\left\{\left|\int_{0}^{x}p_{m}\left(x\right)dx-\int_{0}^{x}p_{n}\left(x\right)dx\right|\right\}$$
(4)$$\delta \left(P_{m},P_{n}\right)\leq \sqrt{\frac{1}{2}D_{KL}(P_{m}\left|\right|P_{n})}$$

It is also important to note that *J* is not an actual distance, while δ is. The PDFs obtained for each of the four stations will be compared pairwise. The lower the values of $D_{KL}$ and δ are, the more alike are the two distributions and the lower the loss of information is between the areas.

#### Other Literature Approach

de Araújo [52] also attempted to assess event duration using stochastic modeling using Equations (Equation 5 to Equation 7). *D* is duration and *H* daily precipitation. $S_{•}$ is the standard deviation of the sample. *j* is a counter index (*j*-th event). χ is a random number such that $\chi^{j}\;\;\in \;\;\left[0,\chi_{max}\right]$. $\chi_{max}$ is calibrated for each watershed. The author proposes that for each event *j*, at least 20 values of $\chi^{j}$ should be drawn. The simulated duration *D* would be the arithmetic average of the 20 produced results.
(5)$$D^{j}=\bar{D}+k^{j}S_{D}$$
(6)$$k^{j}=\frac{H^{j}-\bar{H}}{S_{H}}\chi^{j}$$
(7)$$\frac{\bar{D}-D^{j}}{\bar{H}-H^{j}}=\frac{S_{D}}{S_{H}}\chi^{j}$$

### 2.2. Sediment Yield-PoME—SYPOME Method

de Araújo [26] proposed an entropy-based model for event-based SDR (Equation (Equation 8)) and sediment yield (SSY – Mg km^−1^ yr^−1^). $\bar{\varepsilon }$ (Mg km^−1^ yr^−1^) is the gross erosion obtained, for example, by using the universal soil loss equation—USLE [46], $L_{0}$ the hill slope length (m), $L_{m}$ the maximum sediment travel distance (m), $x_{0}$ the initial position of erosion in the hillslope and λ a Lagrange multiplier.
(8)$$S\;S\;Y=\bar{\varepsilon }\times S\;D\;R=\bar{\varepsilon }\times \frac{e^{\lambda L_{m}}\left(L_{0}-x_{0}\right)\lambda -\left(e^{\lambda \left(L_{0}-x_{0}\right)}-1\right)}{\lambda L_{0}\left(e^{\lambda \left(x_{0}+L_{m}\right)}-1\right)}$$

The SDR is the ratio of sediment yield (SSY) and mobilized sediment ($\bar{\varepsilon }$). The SDR is physically constrained to a closed interval (SDR∈[0,1]), and it can be interpreted as the average probability of a detached particle reaching the river system [26]. The SYPoME model uses as input the duration of the sub-daily precipitation which, in our case, is not known. The MEDRID method can solve this gap, based on daily precipitation.

### 2.3. Monte Carlo and MEDRID-SYPoME Coupling

A Monte Carlo approach was used to adapt the SYPoME model [26] and its output to an interval of possible values of sediment yield associated to a probability function [53]. The results were compared with measured data from seven catchments (Figure 2 and Table 2) and values from the literature model [41].

Using the MEDRID method we can find the probability distribution function (PDF) for the duration–precipitation ratio $\frac{D}{H}$. To model the inherent uncertainty of the duration–precipitation ratio we used the Monte Carlo approach. For each event in the time interval Δt, a large number of random seeds (${\#}_{rand}\in$ [0,1]—Equation (Equation 9)) are generated and used as input in the calibrated PDF to assess the duration (Figure 1).
(9)$${\#}_{rand}=F\left(x\leq \frac{D}{H}\right)=\int_{0}^{D/H}f\left(x\right)dx$$
where *f* is the calibrated PDF according to Table 1 and *F* the associated cumulative distribution function of *x*. Solving Equation (Equation 9) for D/H, with known *H*, we can obtain the rainfall duration for each random seed ${\#}_{rand}$. The set of pairs (D,H) is used as input for the SYPoME model.

### 2.4. Gross Erosion and Siltation Assessment

To estimate gross erosion in the catchments we used the universal soil loss equation (Equation (Equation 10)—[46,54]). A more detailed description of each factor and the values for the study areas can be found in the Appendix A to this paper. Siltation (ΔV) and sediment yield are proportional and related according to Equation (Equation 11).
(10)$$\bar{\varepsilon }=R\;K\;LS\;C\;P$$
(11)$$SSY=\frac{\Delta \;V\;\rho_{s}}{\eta \;A\;\Delta \;t}$$
where ΔV is the volumetric siltation, or the reservoir capacity loss (in m^3^), $\rho_{s}$ is the bulk density of the silted sediment (in Mg m^−3^), η the trap efficiency of the reservoir (using, e.g., the method by [55]), *A* is the catchment area in hectares and Δt the interval of time in analysis.

In order to assess the performance gain by using the MEDRID+SYPoME model, we compared the measured data with empirically based SDR equations [42]. Gaiser et al. [35] found that, for the Brazilian northeast region, the most fit among those equations is the one by Maner [41], (hereafter Equation (Equation 12)). Simplício et al. [56] had the same result for the dry Cerrado region of Gilbués (Figure 2).
(12)$$\mathrm{SDR}=\exp\left[2.943-0.824\;\log_{10}\left(\frac{F_{L}}{F_{R}}\right)\right]$$

$F_{L}$ (m) is the length factor, measured as the maximum distance in the catchment with a straight line from the outlet to the water divide approximately parallel to the main river. $F_{R}$ (m) is the relief factor, calculated as the difference between the outlet altitude and the average altitude of the water divide.

### 2.5. Study Area

We selected seven catchments in three different states of the Brazilian northeast, all under dry conditions (Figure 2) to test the method approach for precipitation downscaling (MEDRID) and the sediment yield assessment model (SYPoME). The catchments vary widely in area and availability of data (number of years in a time series). They also vary in terms of land use and land cover. The characteristics of the studied catchments are listed in Table 2.

The Brazilian northeastern region houses the country’s semiarid region (BSh climate, according to the Köpper Classification—[35]) and the Caatinga Biome. The Caatinga is the largest tropical dry forest in the world and houses the highest endemic genera of all [57,58]. The main economic activities in the region are agriculture (especially maize, beans and soybeans), livestock and fishing [6]. Due to deleterious practices in agriculture and overgrazing, the degraded area surpassed 72,000 km^2^ in the Brazilian Drylands (ca. 8% of its original area—[59]).

As presented in Section 1, the Brazilian northeast region suffers with data scarcity concerning sub-daily rainfall events. Therefore the selection is restricted to the existing (and operating) stations. The stations in Gilbués, Aiuaba and Sumé (Figure 2) were maintained by research groups [13,34,56] and only the station of Sobral is maintained by the Brazilian Water Management Agency (ANA). Those four stations presented consistent measurements over at least two years without gaps. Another constraint for the selection of stations was the proximity to the sediment control equipment. Again, the stations in Gilbués, Aiuaba and Sumé were installed to monitor experimental basins and are inside the catchment areas. The Sobral station was chosen because it is in the Várzea da Volta catchment and is the closest to Acarape under the same climate conditions. For a detailed map of stations in the region please refer to the Appendix A.

Experimental data were used to estimate sediment yield [60]. We used bathymetric assessments from different years of the reservoirs of five catchments (Canabrava, Aiuaba, Várzea da Volta, Acarape and Gilbués) to estimate the total siltation (ΔV—see Equation (Equation 11)). Direct data for sediment yield (SSY) were available at the micro-basins in Sumé, where monitoring is carried out eventwise [13]. Table 2 lists the type and timing of available sediment yield data. For each catchment we obtained the time series of daily rainfall from FUNCEME [61]. Sub-daily measurements are scarce and available for the whole study period only in one station in Gilbués [56] and one in Aiuaba [34], the basins with the shortest and most recent time series. Assuming similar climatic and environmental conditions, we used the data from the Aiuaba station for the analysis of Canabrava, and from Várzea da Volta for Acarape.

## 3. Results

### 3.1. Probability Distribution Functions—MEDRID

Table 3 presents the entropy-based calibrated parameters for B3 (beta distribution), G2 (gamma distribution) and G3 (generalized gamma distribution). Those values were obtained by solving the systems of equations in Table 1. In Figure 3 we present the model evaluators of distributions at the four stations. From the method evaluators we can observe that B3 represents poorly the distribution when compared with the gamma distributions (Figure 3). G3 performs slightly better than G2. From Table 3 we see that the parameter *c* of the generalized gamma does not sufficiently approach the unit (when c=1, the gamma and generalized gamma are equal). The strict two-parameter gamma distribution (G2) does not quite represent the process, but less skewed function G3 does.

Two probability distribution functions were tested for the ratio $I_{30}/H$. The beta distribution (B3) and uniform distribution allow an explicit definition of lower and upper boundaries. For the Sobral, Aiuaba and Gilbués stations the uniform distribution presented much better results, with Nash–Sutcliffe efficiency (NSE) as high as 0.98, while the beta distribution had an efficiency lower than 0.50 (Figure 4). In the Sumé station both B3 and uniform distributions had similar performance with NSE of 0.98 and 0.99, respectively. In this work we used the uniform distribution for the modeling in all regions.

Additionally, using statistical measures, we calculated the information loss resulting from using the PDF calibrated for one region into another (Equations (Equation 2) and (Equation 3)). We compared the four stations with sub-daily data among themselves. The measures (symmetric divergence and Kolmogorov–Smirnov distance) for the variable D/H are given in Table 4.

These measures indicate that there is a considerable difference in the duration–precipitation (D/H) distribution in Gilbués over the other three regions.

Sobral and Sumé also appear to be very similar, despite the distance between them. Located in the Brazilian Semiarid Region, the stations in Sobral, Sumé and Aiuaba are under the same major atmospheric process for rainfall formation (the Inter-Tropical Convergence Zone—ITCZ) and have a similar rainfall regime (more than 70% of the annual precipitation concentrated in three months) and amount (500–600 mm yr^−1^). Gilbués has a higher precipitation rate (1200 mm yr^−1^) and better temporal distribution. Therefore, based on statistical distances (Table 4) and regional characteristics, Sobral and Sumé are most similar and have the lowest information loss when (quality) data from one station are used for the other region. Aiuaba is also similar to Sumé and (especially) to Sobral. Gilbués has particular PDF parameters, with both $D_{KL}$ and δ significantly higher when compared with the other three stations.

### 3.2. Sediment Yield Modeling

Two models were tested to assess sediment yield: a classic model consisting of the multiplication USLE gross erosion ($\bar{\varepsilon }$) and empirically based SDR [41], hereby called model M1, and the proposed MEDRID+SYPoME model (M2).

In Table 5 we present the output of the combination of the MEDRID method and SYPoME model (M2) for the seven study areas. Average modeled sediment yield at the outlet varied between 5 (Aiuaba) and 2346 (Sumé 4) Mg km^−2^ yr^−1^ and SDR between 5.9% (Várzea da Volta) and 29.7% (Gilbués). The outputs for sediment yield and SDR of model M2 passed the normality test [62] and we obtained the confidence interval (*p* = 0.01) using a Gaussian distribution. M1 is a deterministic model, thus it has only one single output, presented in Figure 5.

In Figure 5, we present two plots. Figure 5a shows modeled (M1 and M2) and measured values of siltation rate (siltation rate per unit of area) and Figure 5b the modeled (M1 and M2) values of SDR. The siltation rates generated by our approach (M2) clearly outperform those based on deterministic methods (M1). When assessing average sediment yield for each area, our model also outperforms the deterministic model for all experimental basins, with an error reduction by a factor of at least 2 and as high as 20 (Table 6). In addition, the new methodology (M2: MEDRID+SYPoME) presented better performance evaluators (NSE = 0.96 and RMSE = 608.6 ton km^−2^ yr^−1^) than the conventional (M1) approach (NSE = −4.49 and RMSE = 3286 Mg km^−2^ yr^−1^).

By comparing the values of siltation rate in Figure 5a with land use and land cover (Table 2) we can draw a strong correlation between them. Catchments with preserved vegetation, such as Aiuaba and Sumé 2, have the lowest siltation rate, over two orders of magnitude lower than degraded regions, such as Sumé 4 and Gilbués. Basins with the presence of agriculture (Canabrava, Várzea da Volta and Acarape) presented intermediary rates, although ten times larger than preserved regions.

Figure 5b shows the modeled average SDR (for the whole time series) of the basins obtained by M2 and M1 (Equation (Equation 12)). Considering the area of the basins (Table 2), we can observe a dependency of the SDR to the catchment area. Although M2 also showed a similar tendency, its values of SDR are systematically lower than M1’s. It is interesting to note that for the catchments Canabrava, Acarape and Várzea da Volta there is almost no dispersion of SDR values. This is due to the long time series for those experimental areas. With a long temporal series, the averaging of the SDR of all events tends to a narrow range of values that can be understood as the basin SDR. Additionally, the Maner equation (Equation (Equation 12)) allows values of SDR numerically larger than 100 %, which is inconsistent with the physical interpretation of SDR. Whenever the calculated SDR was larger than the physical limit, the value was limited to 100 %, as is the case of Gilbués.

## 4. Discussion

The complexity of hydrological processes can be better modeled with the help of stochastic approaches [15,22]. Ref. [15] proposed a path for sedimentological models relying on the combination of deterministic and probabilistic models in a so-called third-generation erosion model, to which our method belongs. By introducing stochastic routines and calibrating parameters with the principle of maximum entropy, we extracted from the scarce data more valuable information than by employing deterministic models, and even preserved the local characteristics of each region. The method performed well across a large range of time series and catchment-area scales.

### 4.1. Probability Distribution Functions—MEDRID

In the literature [48,63,64,65] many probability distribution functions are related to precipitation processes (e.g., gamma, power-law, exponential); especially concerning its duration (e.g., gamma, Weibul, lognormal). From Figure 3, we conclude that, although the gamma distribution (G2) does reproduce the D/H ratio, the generalized gamma distribution yields the best results in all study areas. Its better fit to the measured data appears to be related to the high complexity (uncertainty/entropy) involved in rainfall events, when many factors interact simultaneously. In such conditions, a less constrained distribution such as the G3 allows for more flexibility and calibration. With one additional parameter, the function becomes more adaptable to the peculiarities of each region in comparison with G2. This is confirmed by the values obtained for the parameter *c*, which never approximate to one (Table 3). Table 1 shows that a parameter c equal to one reduces a generalized gamma distribution to a conventional one (G2).

Information entropy is a measure of uncertainty [18]. Therefore, the PoME delivers the probability distribution function that maximizes the uncertainty under a set of constraints and avoids unproven assumptions [66]. It can be proven that the uniform distribution, such as the one obtained for $\frac{I_{30}}{H}$, has the highest uncertainty (see [18]).

In the selection of the best distribution using the PoME, additionally to the constraints listed in Table 1, there is an implicit assumption taken: that the data follow a specified distribution (i.e., beta, gamma, uniform, etc.). Silva Filho et al. [67] pointed out that the selected constraints of the PoME have to be relevant to the studied variable and that additional constraints do not necessarily lead to better results. Therefore, as we see in Figure 3 and Figure 4, the narrowest distribution does not necessarily best suit the model. The constraints-quality trade-off problem becomes clear in the modeling of rain intensity (Section 3.1), where the most suitable distribution is the uniform one. Such a result occurs because the unproven implicit constraint (the distribution itself) is shown not to be valid.

The use of a uniform distribution for intensity implies that a stochastic approach is more valid than regression curves, as previously proposed by [68,69,70]. Therefore, in stochastic models, a more realistic approach to be adopted is the uniform distribution, as expressed in Equation (Equation 13). The value of 30-min intensity ($I_{30}$) can vary between 0—in the case of H→0—and 2H (for a precipitation with duration lower than 30 min). Equation (Equation 13) is a general equation and does not depend on calibration. Nevertheless, the implementation of Equation (Equation 13) also requires a Monte Carlo approach, as presented in Section 2.3, with drawing of multiple random seeds (${\#}_{rand}$).
(13)$$I_{30}=\frac{H}{D}+H\left(2-\frac{1}{D}\right){\#}_{rand}\;\;suchthat\;\;\frac{I_{30}}{H}\in \left(0,\;2\right]$$

In terms of regionalization of the MEDRID method, equations calibrated using data from a gauged catchment can be used in ungauged regions, provided that they have similar relief and climatic conditions, thus reducing the loss of information. It is important to note that geographic proximity between the station and the application site is not enough to guarantee better parameter homogeneity and, thus, good model performance. The equations from Sumé and Sobral are remarkably similar, although they are more distant from each other than to Aiuaba. Nevertheless, the conditions of the Aiuaba catchment, which is higher and prone to orographic precipitation, may explain its distinction from the others. Finding the causes of similarities between areas, however, surpasses the scope of this work. Still, from analysis of relief and climate of the studied areas and based on the statistical distances (Table 4), we can build a map of possible factors that influence such similarity (Figure 6). The relative position of each area in Figure 6 is based on geographical location. The connecting lines indicate how similar the areas are to each other.

De Araújo’s (2017, [52]) method of precipitation down-scaling, although simpler, has two problems. Firstly, each precipitation event is processed by the model only once, using an averaged duration as input. This reduces the freedom of the model to simulate extreme cases. The model by de Araújo [52] also tends to represent the process by a linear function, after the averaging (Figure 7). Secondly, the author’s approach assumes a normal distribution of duration and daily precipitation. It is also assumed that both distributions are related by an unknown scaling factor χ (Equation (Equation 7)). None of these assumptions could be confirmed by experimental data.

### 4.2. Sediment Yield Modeling

In all cases the MEDRID+SYPoME model (M2) performed better than the deterministic model (M1) with empirically based SDR. As shown in Figure 5 and in Table 6, the relative error was reduced nine-fold, on average. Except for Várzea da Volta, the average error was 21%, five times smaller than the average error for M1. When also excluding Várzea da Volta, the performance of M1 was similar to values obtained from the literature, see [71]. The Nash–Sutcliffe efficiency of event-wise sediment yield calculated for the catchments of Sumé 2 and 4 (0.52 and 0.47, respectively) can be classified as satisfactory since its efficiency is marginally equal to 0.50 [72]. These are, nonetheless, important results, especially considering the little information required to achieve them. The efficiency of the model for total siltation rate is 0.96 (Table 6); its classification ranges from very good [72] to good [73]. This supports the argument that stationary parameters such as relief (in our temporal analysis scale) play a relevant role for sediment delivery mechanisms [56]; they therefore increase the performance of the model over time.

Both models perform poorly in the assessment of siltation of the Várzea da Volta reservoir (see also [35]). This is mainly caused by the peculiarity of its catchment topography and lithology. As illustrated in Figure 8, the upper (southern) part of the watershed is formed by a plateau ending in a cliff of over 500 meters in depth formed by soil that is prone to erosion (USLE parameter K = 0.032 Mg h MJ^−1^ mm^−1^—[35]). The lower portion of the watershed is mostly flat, and its soil has a higher permeability, promoting an interruption of connectivity and therefore reducing the SDR, similar to the process identified by Medeiros and de Araújo [31] in a flat area upstream of the Benguê Reservoir in north-eastern Brazil. Our model (M2) was not able to describe such behavior, although it significantly reduces the error when compared to the conventional methodology (M1).

One limitation of this study is the use of the universal soil loss equation to assess gross erosion. The USLE does not directly address gully erosion [46]. Nevertheless, gullies may be major sediment sources [74], especially in degraded areas such as Sumé 4 and Gilbués [13,56].

## 5. Conclusions

We have proposed a novel method to downscale duration and intensity of precipitation for erosion modeling based on daily data. The best probability distribution function for the duration–precipitation ratio (D/H) is the generalized gamma distribution (NSE = 0.98). For the ratio $I_{30}/H$, the uniform distribution (NSE = 0.47) performs best. The MEDRID method presents resilience to regionalization, therefore demanding fewer climatological stations to cover a large area and allowing the implementation of the model in regions with data scarcity.

Using the downscaled duration and $I_{30}$ intensity generated by MEDRID, we are able to assess sediment yield with a higher accuracy than conventional USLE and relief-based SDR. The coupling MEDRID+SYPoME model allowed assessment of event-wise sediment yield and presented errors that were six times smaller than the ones from conventional models. The new model (MEDRID+SYPoME), based on the combination of deterministic and entropy-based components, improved substantially the performance of assessment of sediment yield (NSE = 0.96) when compared with deterministic modeling.

Additional studies should be carried out to test and assess the most suited probability distribution families to precipitation data, especially 30-min intensity. Efforts are still necessary to validate the method’s potential concerning regionalization. It is not at all a trivial matter to determine which factors (relief, climate, position, etc.) influence homogeneity between regions, and therefore produce similar PDFs.

The MEDRID method can be used to assess rainfall sub-daily features (duration and 30-min intensity). When coupled as MEDRID+SYPoME, the novel model provides accurate results for sediment yield across a wide range of catchment areas in catchments with areas of different orders of magnitude (from 10^−3^ to 10^+2^ km^2^) and land use.

## Figures and Tables

**Figure 1 entropy-23-01615-f001:**
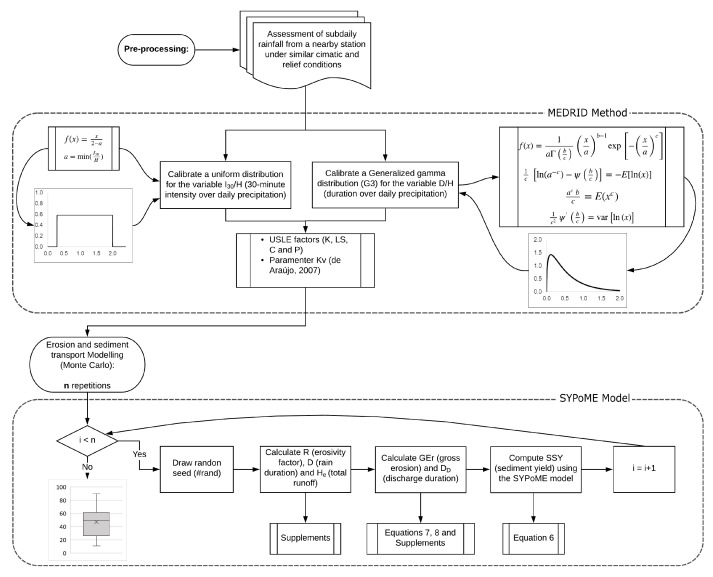
Flowchart of the proposed model. The processing is divided in two main parts, the MEDRID method and the SyPOME model. The two parts are coupled by a Monte Carlo process with multiple random seeds generated.

**Figure 2 entropy-23-01615-f002:**
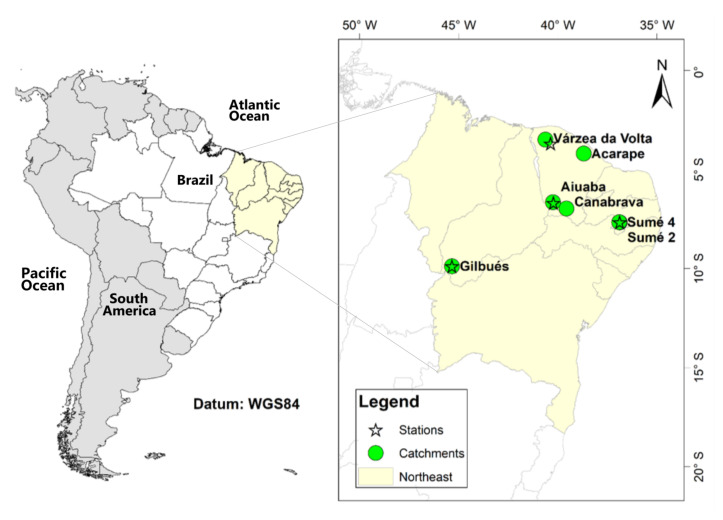
Location of study areas (catchments) and automatic rain gauges. All areas are located in the Brazilian northeast.

**Figure 3 entropy-23-01615-f003:**
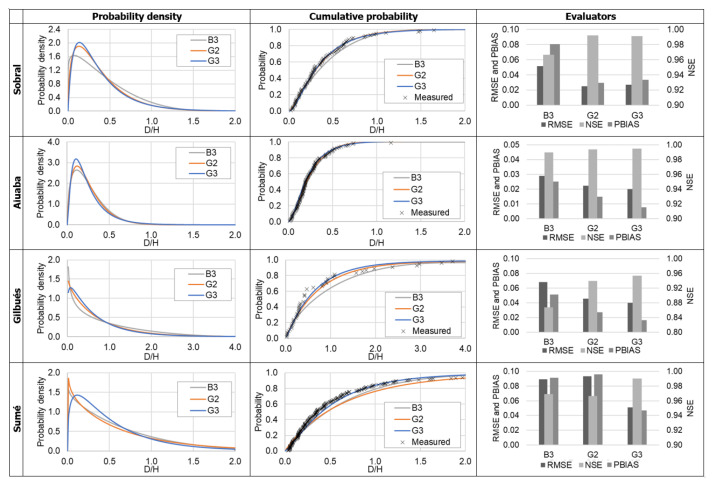
Probability distributions and the performance evaluators for the variable D/H.

**Figure 4 entropy-23-01615-f004:**
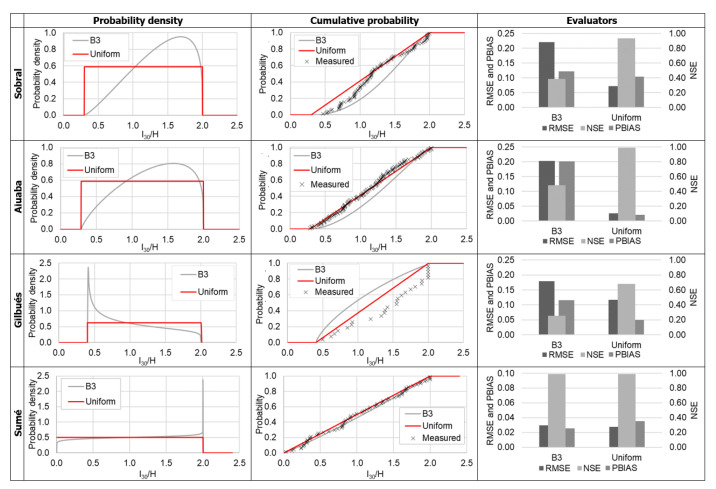
Probability distributions and the performance evaluators for the variable *I_30_/H*.

**Figure 5 entropy-23-01615-f005:**
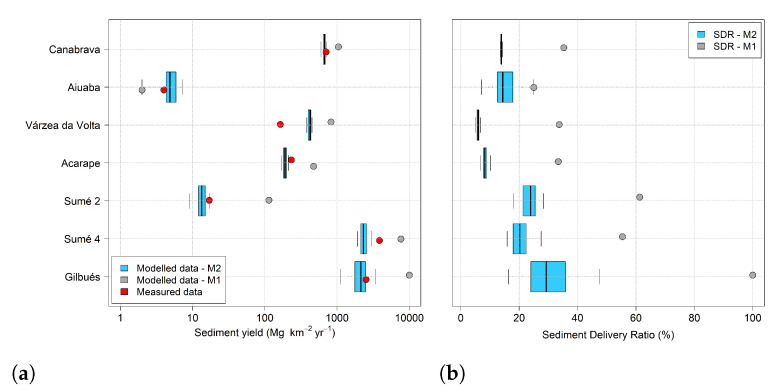
M1 and M2 outputs of (**a**) sediment yield and (**b**) SDR. Red dots in (**a**) indicate the measured values of sediment yield.

**Figure 6 entropy-23-01615-f006:**
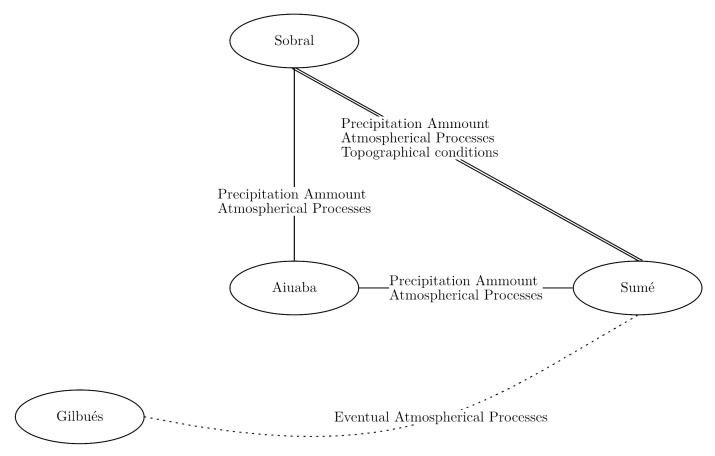
Clustering (connections) of regionalized PDFs and possible influencing factors for the similarities, based on relief and climate conditions. Note that nodes are positions to roughly match the geographical location of each study area (no scale—Figure 2).

**Figure 7 entropy-23-01615-f007:**
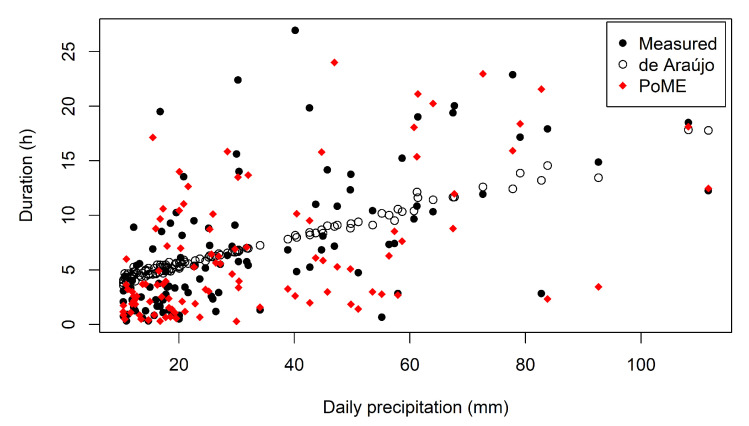
Scatter plot of daily precipitation and duration for Aiuaba. Note that both methods depend on random seeds, therefore the points’ position in the plot is not fixed, but rather an example. Other examples are available in the Appendix A.

**Figure 8 entropy-23-01615-f008:**
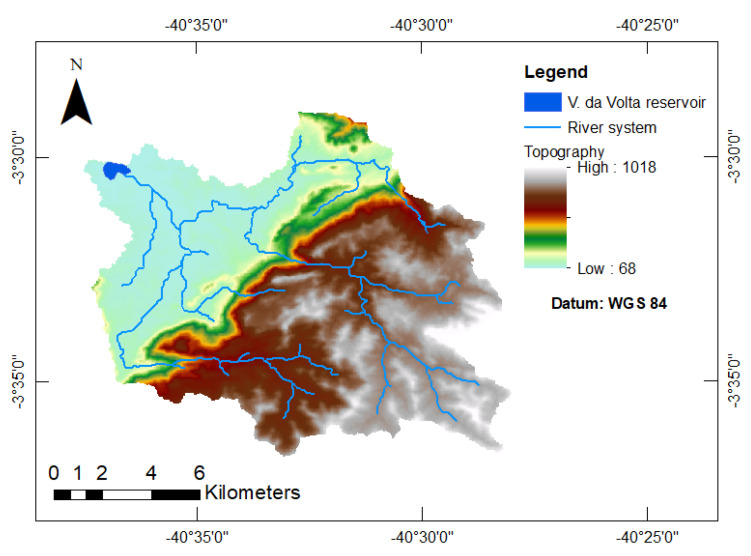
Topography and river system of the Várzea da Volta Catchment area.

**Table 1 entropy-23-01615-t001:** Parameterization of PDFs beta (B3), gamma (G2) and generalized gamma (G3). We present the list of constraints used for each equation and the obtained system after solving with the Lagrange multipliers method.

Equation		B3	G2	G3
PDF		$f\left(x\right)=\frac{\Gamma (a+b)}{\Gamma (a)+\Gamma (b)}{\left(\frac{x}{r}\right)}^{a-1}{\left(1-\frac{x}{r}\right)}^{b-1}$	$f\left(x\right)=\frac{1}{a\Gamma (b)}{\left(\frac{x}{a}\right)}^{b-1}\exp(-\frac{x}{a})$	$f\left(x\right)=\frac{c}{a\Gamma (\frac{b}{c})}{\left(\frac{x}{a}\right)}^{b-1}\exp[-{\left(\frac{x}{a}\right)}^{c}]$
Constraints	*i.*	$\int_{0}^{1}f\left(\frac{x}{r}\right)d\left(\frac{x}{r}\right)=1$	$\int_{0}^{+\infty }f\left(x\right)dx=1$	$\int_{0}^{+\infty }f\left(x\right)dx=1$
	*ii.*	$\int_{0}^{1}\frac{x}{r}f\left(\frac{x}{r}\right)d\left(\frac{x}{r}\right)=E\left[\ln\left(\frac{x}{r}\right)\right]$	$\int_{0}^{+\infty }xf\left(x\right)dx=E\left(x\right)$	$\int_{0}^{+\infty }x^{q}f\left(x\right)dx=E\left(x^{q}\right)$
	*iii.*	$\int_{0}^{1}\ln\left(1-\frac{x}{r}\right)f\left(\frac{x}{r}\right)d\left(\frac{x}{r}\right)=E\left[\ln\left(1-\frac{x}{r}\right)\right]$	$\int_{0}^{+\infty }\ln\left(x\right)f\left(x\right)dx=E\left[\ln\left(x\right)\right]$	$\int_{0}^{+\infty }\ln\left(x\right)f\left(x\right)dx=E\left[\ln\left(x\right)\right]$
System	*i.*	$E\left[\ln\left(\frac{x}{r}\right)\right]=\psi \left(a\right)-\psi \left(a+b\right)$	$ab=\bar{x}$	$\frac{1}{c}\left[\ln\left(a^{-c}\right)-\psi \left(\frac{b}{c}\right)\right]=-E\left[\ln\left(x\right)\right]$
	*ii.*	$\;E\left[\ln\left(1-\frac{x}{r}\right)\right]=\psi \left(b\right)-\psi \left(a+b\right)$	$\psi \left(b\right)-\ln\left(b\right)=E\left[\ln\left(x\right)\right]-\ln\left(\bar{x}\right)$	$\frac{a^{c}\;b}{c}=E\left(x^{c}\right)$
	*iii.*			$\frac{1}{c^{2}}\psi^{\prime }\left(\frac{b}{c}\right)=\mathrm{var}\left[\ln\left(x\right)\right]$

**Table 2 entropy-23-01615-t002:** Study area information. Lines with same color indicate areas that share automatic rain gauge data.

Basin	Area (km^2^)	Control System	Land Use	Location	Catchment Position		Bathymetry		Time Series	Automatic Weather Station				
										Name	Position		Recording	
					Lon	Lat	First ^a^	Second			Lon	Lat	Start	End
Canabrava	2.9	Reservoir	Agriculture and open range cattle raising	Ceará	39.56 W	6.97 S	1944	2000	57	Aiuaba	40.22 W	6.69 S	2004	2014
Aiuaba	11.53	Reservoir	Conservation area with native vegetation (Caatinga)	Ceará	40.24 W	6.65 S	2003	2009	7	Aiuaba	40.22 W	6.69 S	2004	2014
Várzea da Volta	155	Reservoir	Agriculture and open range cattle raising	Ceará	40.62 W	3.50 S	1917	1997	81	Sobral	40.36 W	3.69 S	2017	2019
Acarape	208	Reservoir	Agriculture and open range cattle raising	Ceará	38.69 W	4.20 S	1924	1999	74	Sobral	40.36 W	3.69 S	2017	2019
Sumé 2	0.0107	Sediment load	Experimental area—preserved vegetation	Paraíba	36.88 W	7.67 S	-		10	Sumé	36.88 W	7.67 S	1982	1991
Sumé 4	0.0048	Sediment load	Experimental area—degraded land without vegetation	Paraíba	36.9 W	7.66 S	-		10	Sumé	36.88 W	7.67 S	1982	1991
Gilbués	0.0004	Check dam	Abandoned land under desertification process without vegetation	Piauí	45.34 W	9.88 S	2018	2019	1	Gilbués	45.34 W	9.88 S	2018	2019

^a^ The first bathymetry corresponds to the topography in the year of construction.

**Table 3 entropy-23-01615-t003:** Equation parameters for the D/H distribution. *a*, *b* and *c* are the parameters as described in Table 1. The data used to calibrate the parameters are available in the Appendix A.

	B3		G2		G3		
	*a*	*b*	*a*	*b*	*a*	*b*	*c*
**Sobral**	1.124	4.316	0.250	1.525	0.066	2.114	0.678
**Aiuaba**	1.584	10.686	0.138	1.855	0.004	3.306	0.488
**Gilbués**	0.696	2.691	0.777	0.953	0.390	2.099	0.812
**Sumé**	0.955	5.398	0.740	0.911	0.269	1.410	0.818

**Table 4 entropy-23-01615-t004:** Values of symmetric divergence and Kolmogorov–Smirnov distance for the generalized gamma distribution of D/H. The higher the value, the greater the difference between the probability distributions.

(a) Symmetric Divergence				
	Sobral	Aiuaba	Gilbués	Sumé
Sobral	0	0.198	1.210	0.097
Aiuaba	0.198	0	2.494	0.536
Gilbués	1.210	2.494	0	0594
Sumé	0.097	0.536	0.594	0
**(b) Kolmogorov–Smirnov Distance**				
	Sobral	Aiuaba	Gilbués	Sumé
Sobral	0	0.242	0.550	0.152
Aiuaba	0.242	0	0.719	0.365
Gilbués	0.550	0.719	0	0.404
Sumé	0.152	0.365	0.404	0

**Table 5 entropy-23-01615-t005:** Modeled values (M2) of sediment yield and SDR for the study areas. The values are shown in terms of average (μ), standard deviation (σ) and coefficient of variation (CV). Confidence intervals (CIs) of the average calculated for p = 0.01.

Basin	Sediment yield				SDR			
	(Mg km^−2^ yr^−1^ )				(%)			
	μ	σ	CV	CI	μ	σ	CV	CI
**Canabrava**	664.5	24.9	4%	12.5	13.9	0.2	1.4%	1.04
**Aiuaba**	5.0	1.2	25%	0.6	14.8	4.2	28.4%	2.12
**Várzea da Volta**	418.2	20.2	5%	10.1	5.9	0.4	7.3%	0.22
**Acarape**	189.5	9.1	5%	3.1	8.3	0.7	8.1%	0.23
**Sumé 2**	13.1	1.8	14%	0.9	23.5	2.6	11.1%	1.32
**Sumé 4**	2345.6	264.1	11%	132.9	20.4	3.0	14.6%	1.50
**Gilbués**	2141.7	540.5	25%	272.0	29.7	8.9	29.9%	4.47

**Table 6 entropy-23-01615-t006:** Measured and modeled values of siltation rate (Mg km^−2^ yr^−1^). M1 represents the classic model using empirically based SDR (Maner, Equation (Equation 12)) and M2 the proposed MEDRID+SYPoME model. Summaru line numbers are in boldface.

Name	Brune Coefficient	Sediment Yield (Mg km^−2^ yr^−1^)			Relative Error (%)	
		Measured	Modeled	Modeled	Modeled	Modeled
			M1	M2	M1	M2
Canabrava	0.98	704	1042	664	*48.0%*	*−5.6%*
Aiuaba	1.00	4	2	5	*−50.0%*	*27.5%*
Várzea da Volta	0.95	164	824	418	*402.3%*	*155.1%*
Acarape	0.98	233	473	191	*102.9%*	*−18.1%*
Sumé 2	1.00	17	114	13	*570.6%*	*−21.8%*
Sumé 4	1.00	3857	7644	2314	*98.2%*	*−40.0%*
Gilbués	1.00	2518	10305	2142	*309.3%*	*−14.9%*
**NSE**			**−4.49**	**0.96**		

## Data Availability

Code and data are available at https://github.com/pedroalencar1/MEDRID-SyPOME, accessed on 20 November 2021.

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
