# Peer review of "Entropy-Based Temporal Downscaling of Precipitation as Tool for Sediment Delivery Ratio Assessment"

_entropy, 2021, doi:10.3390/e23121615_

Round 1
Reviewer 1 Report
The research topic is interesting and relevant.
Unfortunately, the level of self-citation is too high - 14 (numbers 6, 25, 30, 33, 34, 36, 38, 39, 40, 54, 58, 62, 70, 72) references out of 77. This is 18.2% when the maximum level is 15%. I recommend replacing some of the references.
Author Response
We thank the Reviewer #1 for their valuable time and work spent in our manuscript.
We agree with the comments and proceeded with the necessary reviews. Redundant references were removed and self-citations removed when necessary to achieve the recommended level.

Reviewer 2 Report
The manuscript deals with hydrological modelling through the principle of maximum entropy. The Introduction and the literature review are joint together in the first section. The Materials and Methods chapter offers an overview of the techniques applied in the analysis. It is quite detailed. In the same light, the analytical outcome is introduced. On the b side of Figure 5, Gilbués reaches through its highest value a level of 100 % of sediment delivery ratio. Is it the right value? Or is it only a very strange outlier? Why did you choose uniform probability distribution in order to model the obtained values? The Discussion section brings the very interesting thoughts. They are related also to the other studies. It is considerably beneficial for reader. Also, the contents of the discussion are rich. The Conclusions summarise the obtained findings finally.
There are some formal mistakes and grammar errors, but they do not lower quality of the manuscript – for instance:
- a missing comma on line 4: “In this work we propose…” instead of “In this work, we propose…”;
- a missing comma on line 276: “In Figure 5 we present…” instead of “In Figure 5, we present…”;
- a missing space on line 362: “…distances(Table 4)…” instead of “…distances (Table 4)…”.
Author Response
We thank the Reviewer #2 for their valuable time and work spent in our manuscript.
We agree with the comments and proceeded with the necessary reviews. Grammar issues and typos have been fixed.
On figure 5b, the value for Gilbués is indeed correct. The grey dots are not the outliers of the box plot, but the value of Sediment Delivery Ratio obtained by the equation of Manner (M1 - Eq. 10). The Equation of Manner allows the SDR value to be even larger than 100 % and this was the case for Gilbués. However, given the physical interpretation of what SDR is, it is physically inconsistent to have values above 100%, therefore, in our code we set a rule of [if SDR > 100, then SDR = 100]. A sentence was added to the text to clarify the value.
Regarding the use of the uniform distribution, we tested multiple possible PDFs to model the 30-minute intensity and the duration. For the 30-minute intensity, two distributions presented a fair performance: B3 (or beta distribution) and the uniform, having the latter presented the best results. Other distributions should be explored with larger sets of data to validate or refute this finding. It is also important to highlight that changing the distribution of I30 does not change the proposed methodology.
